# Epigenetics, Nutrition, and the Brain: Improving Mental Health through Diet

**DOI:** 10.3390/ijms25074036

**Published:** 2024-04-04

**Authors:** Rola A. Bekdash

**Affiliations:** Department of Biological Sciences, Rutgers University, Newark, NJ 07102, USA; rbekdash@newark.rutgers.edu

**Keywords:** brain, epigenetics, gene-environment interactions, mental health, metabolism, micronutrients, methylation, methyl-donors, nutrition

## Abstract

The relationship between nutrition and brain health is intricate. Studies suggest that nutrients during early life impact not only human physiology but also mental health. Although the exact molecular mechanisms that depict this relationship remain unclear, there are indications that environmental factors such as eating, lifestyle habits, stress, and physical activity, influence our genes and modulate their function by epigenetic mechanisms to shape mental health outcomes. Epigenetic mechanisms act as crucial link between genes and environmental influences, proving that non-genetic factors could have enduring effects on the epigenome and influence health trajectories. We review studies that demonstrated an epigenetic mechanism of action of nutrition on mental health, focusing on the role of specific micronutrients during critical stages of brain development. The methyl-donor micronutrients of the one-carbon metabolism, such as choline, betaine, methionine, folic acid, VitB6 and VitB12 play critical roles in various physiological processes, including DNA and histone methylation. These micronutrients have been shown to alter gene function and susceptibility to diseases including mental health and metabolic disorders. Understanding how micronutrients influence metabolic genes in humans can lead to the implementation of early nutritional interventions to reduce the risk of developing metabolic and mental health disorders later in life.

## 1. Introduction

Are we really what we eat? Do our dietary choices play a role in our susceptibility to diseases, including mental health disorders? Genetic and environmental factors both play critical roles in predisposing individuals to developing mental health conditions later in life [1,2]. The prevalence of mental health disorders is increasing, significantly impacting society, and affected individuals. According to the World Health Organization, as of 2019, approximately one in every eight people, or 970 million people globally, were suffering from mental health conditions [3]. In 2021, the National Institute of Mental Health estimated that more than one in five adults in the United States lives with a mental condition [4]. Obesity represents another growing public health issue globally, further illustrating the complex relationship between diet, health, and disease. World Health Organization data show a significant increase in obesity rates since 1990, with adult obesity doubling and adolescent obesity quadrupling, a troubling trend that mirrors the rise in mental health disorders. In 2022, it was estimated that one in eight people worldwide were living with obesity, and 37 million children under the age of five were overweight [5,6]. These statistics are alarming and emphasize the urgent action to address and solve these health challenges. The underlying mechanisms of these health conditions are complex and multifactorial. Recent advances in epigenetics and neuroscience reveal that mental health conditions are not only caused by genetic factors but also are linked to environmental factors. These include diet quality and quantity, stress exposure and coping mechanisms, physical activity levels, early life experiences, and socioeconomic status [7,8,9]. Exposure to these factors during early life can have lasting effects not only on physiology but also on mental well-being [10,11].

This review highlights the complex relationship between nutrition and epigenetic mechanisms in mental health. Several mechanisms have been proposed to understand the effects of nutrition during early life on brain optimal function and chemistry across the lifespan. Epigenetic mechanisms such as DNA methylation, histone modifications and the role of microRNAs are considered plausible mechanisms that influence the onset of various diseases including mental health disorders [12,13,14]. The early stages of development are particularly sensitive to environmental influences and are characterized by substantial changes in brain plasticity. These changes can significantly reshape the brain’s epigenetic landscape, especially during its critical developmental periods. We will address the role of micronutrients that are synthesized in small amounts in the body and that act as methyl-donors in the one-carbon metabolism such as choline, betaine, folate, methionine, VitB6 and VitB12 and modulate the expression of key genes such as stress-related genes and metabolic genes by epigenetic mechanisms during early life. These methyl-donor micronutrients are involved in complex biochemical pathways of one-carbon metabolism, acting as intermediates or cofactors.

Understanding nutrient-gene interactions during the early stages of life is crucial for preventing early childhood obesity and mental health problems. With the guidance of dietitians and health professionals, insights into these interactions can inform preventive strategies and foster the development of early personalized dietary and lifestyle habits that promote mental well-being throughout life. This review first discusses the role of methyl-donor micronutrients in the normal functioning of the one-carbon metabolism. It then summarizes evidence from animal studies and limited human research on the effects of these micronutrients on brain function, metabolic markers, and mental health, highlighting the potential of early nutritional interventions to support mental well-being, showcasing, and advocating for the necessity of integrating nutritional science with personalized health care from a young age.

## 2. One-Carbon Metabolism, Methyl-Donor Micronutrients and Epigenetic Mechanisms

The genome is relatively static throughout life, but the epigenome is constantly changing in response to early life experiences and exposure to environmental factors. These changes are regulated by epigenetic mechanisms such as DNA methylation or histone modifications. These mechanisms are mediated by the activity of enzymes known as “writers” or “erasers “of epigenetic marks. This means that these mechanisms are reversible, making the components of the epigenetic machinery potential drug targets [15]. Moreover, the activity of these enzymes is dependent on the availability of specific metabolites indicating that epigenetic mechanisms are responsive to metabolic changes [16]. What is epigenetics? The term “epigenetics” was first described by Conrad Waddington in 1942 as the “branch of biology which studies the causal interactions between genes and their products, which bring the phenotype into being”. Later the definition was fine tuned to describe epigenetics as heritable changes in gene expression and function without changes in the sequence of the gene in response to environmental factors or early life experiences [17]. The accumulation of epigenetic changes over time could alter an individual’s health trajectories in a positive or negative manner [18]. Nutrients are considered epigenetic regulators that interact with our epigenome and alter how our genes will function throughout life. With a better comprehension of this intricate interplay, we can explore alternative early non-pharmaceutical interventions as alternative solutions to improve people’s quality of life.

We have different types of nutrients, macronutrients, and micronutrients, which are essential for normal physiology and health. In this review, we will focus on the role of specific micronutrients that act as methyl-donors in the one-carbon metabolism and can modulate the epigenome by epigenetic mechanisms [19]. These methyl-donor micronutrients such as choline, folate, methionine, betaine, VitB6 and VitB12 contribute to the formation of the universal methyl-donor, S-adenosylmethionine (SAM). SAM is required for DNA and histone methylation, two epigenetic mechanisms that regulate mammalian gene expression and are needed for neurodevelopment processes [20,21]. SAM also plays a role in other biological processes. The one-carbon metabolism consists of several chemical reactions of the folate cycle, methionine cycle, and transsulfuration pathway. These reactions support DNA, proteins and lipid synthesis, production of neurotransmitters and supply methyl groups via SAM to enzymes, such as DNA methyltransferases (DNMTs) or histone methyltransferases (HMTs/KMTs), for the modulation of gene expression by methylation (Figure 1).

The functionality of the one-carbon metabolism is critical during early development. Dysregulation in the expression of cofactors or components of one-carbon metabolism can have detrimental effects on fetal growth and fetal neurodevelopment [23]. Thus, there is a relationship between nutrition and brain health [24]. How do these micronutrients affect neurodevelopment and what is the role of epigenetic mechanisms in this context?

Early stages of development such as prenatal, postnatal and adolescence are quite vulnerable to environmental factors such as exposure to stressors, poor diet, drugs of abuse, etc. Studies have shown that early exposure to these factors can have long-lasting effects on health trajectories including adverse neurodevelopmental outcomes that could affect an individual’s cognitive functions and behavioral responses [25,26]. It has been proposed that these effects induced by early life exposure to environmental influences are caused by epigenetic mechanisms [27]. Methyl-donor micronutrients play a pivotal role in the functioning of the one-carbon metabolism, indicating that its functionality is dependent on the nutritional status and availability of these micronutrients. As shown in Figure 1, one-carbon metabolism does not only involve folate and methionine cycles, but also shows how these micronutrients act as cofactors or intermediates in several chemical reactions that contribute to SAM formation, which is essential for methylation of histones, DNA, RNA, and other proteins. Other micronutrients like betaine and choline are critical for production of phospholipids such as phosphatidylcholine for cellular membranes and neurotransmitters such as acetylcholine, involved in cognitive functions like attention and memory. Folate and methionine provide via SAM formation the methyl group for enzymes that methylate DNA or histones such as DNA methyltransferases (DNMTs) or histone methyltransferases (HMTs/KMTs) to modulate neuronal gene expression. After the donation of the methyl group, SAM is converted to S-adenosylhomocysteine (SAH). Homocysteine generated by the activity of S-adenosylhomocysteine hydrolase (SAHH) on SAH can then enter the transsulfuration pathway to synthesize glutathione and prevent the accumulation of reactive oxygen species in the brain [28].

Methyl-donor micronutrients are essential for normal brain development. A deficiency in their levels during early life has been linked to neural tube defects, altered behavior, and altered cognitive functions [29,30]. Here we summarize some of the literature that demonstrated a link between methyl-donor micronutrients, brain function and epigenetic changes. For example, gestational choline deficiency in mice decreased global methylation in the hippocampus with a decrease in the methylation of a key regulator of the cell cycle (Cdkn3) and increased its expression in the fetal brain. This suggests that choline deficiency during the critical period of brain development can alter gene expression and brain development by methylation [31]. A study investigated the effects of maternal choline supplementation in iron-deficient rats on offspring health. Choline supplementation from GD11–GD18 mitigated the effects of iron deficiency on the expression of hippocampal genes associated with anxiety, autism, and schizophrenia in adult offspring. Although choline altered the hippocampal transcriptome, this alteration was not demonstrated to be caused by epigenetic mechanisms [32]. Another study reported that maternal supplementation with the methyl-donor folate during the late gestational period (GD13-GD20) normalized the levels of two microRNAs, let-7a and miR-34, and reversed the developmental defects that were induced by maternal folate deficiency at GD20. By normalizing the levels of these two microRNAs, folate supplementation normalized the expression of their target genes that are related to embryonic development, cell migration, axon guidance and vesicular trafficking [33].

In the context of metabolism, the supplementation of methyl-donors to a high fat diet mouse model caused global hypomethylation in the reward brain regions such as the ventral tegmental area (VTA), prefrontal cortex (PFC) and nucleus accumbens (NAc) of offspring. This change correlated with increased locomotor activity in the offspring of methyl-donor supplemented dams compared to the offspring of non-supplemented dams. Moreover, female offspring of methyl-donor supplemented dams showed an increased metabolic rate. Both male and female offspring of methyl-donor supplemented dams showed a normalized preference to fat consumption. Although, this study showed an alteration in the expression of the dopamine transporter (DAT) and the µ-opioid receptor (MOR) expression in specific reward brain regions in offspring of methyl-donor supplemented dams, this change was not caused by methylation [34].

In a genetic rat model of high and low anxiety and depressive-like behavior, epigenetic changes were detected between the two types. The adult rat with a high anxiety phenotype showed 793 differentially methylated sites (DMRs) in the amygdala. Some of these sites were hypermethylated compared to the rat with a low anxiety behavior phenotype. Some of these hypermethylated genes were implicated in mood disorders and emotional disturbances such as NMDA glutamate receptor subunit NR2B (*Grin 2b*) and the presynaptic genes that regulate synaptic function such as *Pclo* and *Stx1b*. Increasing dietary methyl-donor content in the high anxiety group improved anxiety and depressive-like behavioral phenotypes which were demonstrated as decreased in immobility in the forced swim test [35].

## 3. The Role of Key Methyl-Donor Micronutrients and Epigenetic Mechanisms in Metabolic and Mental Health Disorders

Several studies conducted in rodents showed that maternal methyl-donor intake during critical stages of development could alter metabolic genes in the brain of offspring by epigenetic mechanisms [36,37,38]. This suggests that an optimal level of these micronutrients is essential to normalize metabolism and hence mental well-being later in life. Obesity is a main public health problem worldwide and is linked to the development of mental health problems. It has also been suggested that the non-genetic basis of obesity could be attributed to environmental factors such as maternal nutrition that could cause fetal programming and alter its future health trajectory by epigenetic mechanisms [39]. Although the relationship between nutrients and brain health is complex, correlational, and multifactorial, there is evidence that the quality and quantity of nutrients consumed in diet can impact mood and mental health [8,40]. Among these nutrients, the methyl-donor micronutrients are considered neuroprotectants [41,42,43,44]. They have been shown to affect several aspects of brain development by influencing the epigenetic modification of genes that play a role in this process [45]. In fact, the dysregulation of the brain epigenome by environmental factors during embryonic development can cause “fetal programming” with lasting effects on brain functioning and has been linked to many neurological disorders including mental health disorders [18,46,47,48].

Both genetic and environmental factors can affect the structural and functional organization of the brain, contributing to the uniqueness of each individual’s physiology and behavior [1,49,50]. This raises the question of how optimal intake of methyl-donor micronutrients during early life might affect our vulnerability to metabolic or mental health conditions later in life. It prompts further inquiry into whether sustained healthy dietary changes and lifestyle habits can lead to a longer healthier life. Brain development is a complex multistage process that is shaped by genetic factors but also by early life experiences and environmental factors. Given the role of methyl-donor micronutrients in several biochemical processes of one-carbon metabolism such as the production of neurotransmitters and the methylation of DNA and histones, it is not surprising that an imbalance in the levels of these micronutrients can influence the brain with profound effects on the developing and the aging brain [51]. Human studies showed that the maternal intake of methyl-donor micronutrients improved cognitive and behavioral outcomes in offspring [30]. Here, we present findings on the effects of these micronutrients in rodents and in humans on health with a focus on their effects on development of metabolic and mental disorders.

### 3.1. Folate

Folate is derived from diets such as leafy vegetables, fruits, and grains. It is a methyl-donor micronutrient that plays a role in nucleotide synthesis (Figure 1). It is needed for brain development and its deficiency has been linked to neural tube defects [29]. Folate deficiency during the gestational period GD11-GD17 decreased neuronal stem cell (NSC) differentiation and increased neuronal apoptosis in the fetal mouse brain whereas choline supplementation reversed some of these effects [52,53]. In the context of metabolism and energy balance circuitry, animal studies showed that a maternal diet deficient in folate and VitB12 and low in choline during the gestational and lactation period altered the levels of key metabolic genes in the rat offspring. For example, it caused an elevation of plasma peptide YY and increased hypothalamic *Pomc* gene expression. These changes correlated with abnormal levels of important metabolic markers such as leptin, ghrelin, and insulin during a critical period of hypothalamic development. Perinatal supplementation with folic acid mitigated some of these observed effects in pups at postnatal day (PD21) as the hypothalamic expression of *Npy*, *Pomc*, *Lep* and *InsR* were normalized [54]. It was demonstrated in another study that feeding the rat offspring with a diet high in folate changed *Pomc* gene expression by causing a decrease in the methylation of its promoter. This change positively correlated with offspring glucose response to a glucose load [55].

Neonatal isolation from the mother in rats for 90 min daily from PD1-PD11 resulted in autistic-like behaviors in offspring such as a deficit in social engagement, increased repetitive behaviors and anxiety-like behaviors with increased levels of oxidative stress markers (MAD and H_2_O_2_), mild inflammation, a decreased granule cell layer in the frontal cortex with the hypomethylation of the brain derived neurotrophic factor (BDNF), important for brain development, and the glial fibrillary acidic protein (GFAP), important for proper functioning of glia in supporting neurons, in the frontal cortex. The hypomethylation of BDNF resulted in its overexpression, a finding seen in autism spectrum disorder. Daily folic acid supplementation of this neonatal isolation rat model reversed the changes seen in offspring. This suggests the potential use of folic acid, a potent methyl-donor and neuroprotectant, as a viable intervention method to alleviate autistic and anxiety-like behaviors in offspring [56].

Human studies showed that high folate and low VitB12 concentrations during pregnancy are associated with the development of insulin resistance and adiposity in children. This suggests that it is crucial to maintain optimal levels of micronutrients intake, avoiding both excess and deficiency [57,58,59,60]. In a recent study, researchers investigated the impact of one-carbon metabolism micronutrients intake by diet recalls such as choline, folate, and betaine on fetal growth and stress response. The study analyzed changes in fetal DNA methylation in pregnant women with gestational diabetes and those without gestational diabetes. Maternal choline intake showed a positive correlation with cord blood corticotropin-releasing hormone (CRH) methylation and an inverse correlation with cortisol levels in both groups, indicating a potential normalization of stress response through choline. Conversely, an elevated maternal intake of betaine and folate led to the reduced methylation of Insulin Growth Factor 2 in both cord blood and placenta. Additionally, a noteworthy inverse correlation was observed between the maternal intake of betaine and infants’ birthweight. This study once again underscores the association between early life micronutrient exposure and key parameters influencing the metabolism and mental well-being of offspring [61].

In a clinical study conducted in Spain, obese female adult subjects with insufficient folate intake, falling below the Spanish recommended dietary levels due to unhealthy eating habits, was associated with insulin resistance. Insufficient folate intake was also linked to a reduction in the methylation of metabolic-related genes that regulate glucose homeostasis and energy balance such as the calcium/calmodulin-dependent protein kinase kinase 2 (CAMKK2) in white blood cells, with no changes observed in the methylation of other metabolic markers [62].

### 3.2. Choline

Choline is an essential nutrient that is derived from diets such as meat, fish, some beans, milk, and eggs. It is a main methyl-donor micronutrient essential for brain development during critical stages of embryonic development [63]. Choline contributes to the formation of the neurotransmitter acetylcholine which is the main component of phosphatidylcholine and sphingomyelin, two main phospholipids that maintain the structural and functional integrity of cellular membranes (Figure 1). Via betaine, choline can form SAM, thus making choline not only a player in cholinergic neurotransmission at nicotinic receptors but also in gene methylation [64,65,66,67,68]. In the context of metabolism, choline has a role in regulating neuronal circuitry related to food intake. For example, the rat offspring of dams fed a high choline diet (2.5-fold) during pregnancy had a higher hypothalamic expression of neuropeptide y (*Npy*) in the arcuate nucleus at birth, a neuropeptide that plays a role in appetite. The offspring showed increased food intake and body weight gain postweaning, suggesting that choline programs the hypothalamic circuitry that regulates energy balance and the later metabolic phenotype of offspring [69].

There are many different types of mental health disorders such as anxiety, depression, bipolar disorder, post-traumatic disorder, schizophrenia, and neurodevelopmental disorders. Some of these disorders are attributed to genetic factors, others to environmental influences, and yet others arising from a combination of both genetic and environmental factors. Is there a relationship between choline and mental health? What do we know about this relationship? A low intake of choline during the gestational period GD12-GD17 in rodents disrupted fetal progenitor cell proliferation, differentiation, and migration in the subventricular zone of the fetal hippocampus and this was linked to changes in the methylation of key neuronal genes by epigenetic mechanisms [70]. There is a correlation between the prenatal intake of choline and the development of schizophrenia later in life, since cholinergic neurotransmission at alpha-7-nicotinic receptors and a polymorphism in its gene (*CHRNA7*) during the fetal period is one of the physiological inhibitory mechanisms that is found to be dysregulated in this complex disorder. This suggests that an optimal intake of choline during pregnancy is required to mitigate or decrease the risk of developing this mental illness later in life, although there is no proven causal relationship between choline and schizophrenia [71]. The *PEMT* gene that encodes for the enzyme that converts phosphatidylethanolamine to phosphatidylcholine has been linked to development of schizophrenia in Asians due to the identification of three single nucleotide polymorphisms (SNPs) at the PEMT locus [72].

The Hordaland Health Study revealed a link between low levels of choline in the plasma of adults and the onset of anxiety, suggesting the importance of choline in cognitive functions [73]. In a study involving Black African American pregnant women, lower plasma choline levels at 16-week gestation were associated with developmental issues in offspring, specifically affecting cerebral P50 inhibition. This inhibition is crucial for filtering redundant and irrelevant information in newborns examined in this study [74]. Choline’s impact on the development of cerebral P50 inhibition is well-established, a process essential for the postnatal conversion of GABA-sensitive chloride channels into inhibitory ones. It is noteworthy that individuals with schizophrenia exhibit deficiencies or abnormalities in sensory gating processes, leading to incomplete conversion [75].

An imbalance in the levels of prenatal choline and folate on the development of metabolic disorders in offspring has been reported in several animal studies. There is also a link between the development of metabolic disorders and the emergence of mental health disorders, thus suggesting that the optimal levels of these micronutrients, folate, and choline, during early life can cause metabolic programming in offspring and shape health trajectory in adulthood [76]. The programming of the epigenome by these micronutrients has been shown to be mediated by epigenetic mechanisms because they both contribute to the formation of SAM [76]. For example, gestational supplementation with a high methyl vitamin diet (tenfold folate, VitB12 and VitB6) in a Wistar rat model led to the altered methylation of proopiomelanocortin (*Pomc*) and serotonin receptor *5-Htr2a* genes in the adult hypothalamus. This resulted in an elevated *Pomc* expression, although not reaching control levels, and reduced *5-Htr2a* gene expression, accompanied by leptin and insulin resistance. These findings suggest an association between excessive micronutrient intake, folate in this case, and the development of metabolic syndrome in rats during adulthood. It is important to note that proopiomelanocortin and serotonergic pathways play a pivotal role in regulating food intake in the hypothalamus [36].

Another study investigated the intergenerational effects of paternal supplementation of a folate–methionine–choline deficient diet (FMCD) in F0 male mice, examining its impact on the behavior and expression of memory-related genes in the male offspring (F1-FMCD mice). The deficiency of these micronutrients resulted in altered behavior in the offspring during various behavioral tests. These offspring showed anxiety-like behavior, and a non-significant increase in the methylation of a critical memory-related gene (*PP1c*) in the hippocampus. This data correlated with a significant reduction in freezing, indicating an impairment in the consolidation of conditioned fear memory in F1 male mice [77]. In the context of stress impact on behavior, a study investigated the effects of supplementing SAM to the rat diet prior to an acute stress test, the forced swimming test, on methylation and the expression of early induction genes (EIGs) such as *Fos* and early growth response element 1 (*Egr-1*) in dentate gyrus and hence on immobility behavior. SAM led to increased methylation in CpG sites along these genes and decreased their expression, along with elevated levels of the de novo methyltransferase Dnmt3a. These molecular changes were linked to a decreased immobility response after 24 h of SAM supplementation, indicating a diminished stress response [78].

### 3.3. VitB

VitB12 is a micronutrient that is also derived from food such as meat, fish, dairy products, and eggs. It contributes to one-carbon metabolism functionality by facilitating the formation of methionine from homocysteine. This in turn leads to SAM formation, which is essential for the synthesis of neurotransmitters, phospholipids, and nucleotides. In addition, VitB12 plays a key role in regulating methylation reactions that influence gene expression (Figure 1). There is a link between VitB12 deficiency and neurological impairments as its depletion can result in SAM depletion, SAH excess and the inhibition of methyltransferases, resulting in significant molecular changes including changes in the methylation of DNA and histones and changes in gene expression [79].

VitB12 plays an important role in fetal neurodevelopment [80]. Its deficiency has been linked to oxidative stress [81], neuropathy [82], and depression [83]. Focusing on the role of VitB12 in the nervous system and modulation of gene expression by epigenetic mechanisms, here we summarize some findings in the context of VitB12 and its potential role in mitigating the symptoms of mental health such as depression. Depression is a multifaceted global health condition that warrants increased attention for more effective outcomes in affected individuals. VitB12, along with VitB6 and folate, plays a role in neuronal functions, influencing multiple reactions and pathways, including those associated with depression. This suggests that the supplementation of VitB may help alleviate the symptoms of this disorder [83,84]. A study conducted in the US adult population revealed that an increased intake of VitB12 and VitB6 correlated with a reduced likelihood of developing depressive symptoms over a 12-year follow-up period [85]. Additionally, an association was found between folate and VitB12 levels and schizophrenia. Elevated homocysteine levels, stemming from reduced SAM levels, have been reported in schizophrenia. In a limited three-month human study, supplementation with folic acid and VitB12 in schizophrenic patients with high plasma levels of homocysteine showed improvement in symptoms, as indicated by certain neuropsychological tests [86].

A study conducted in Mexican American children aged 8–15 years revealed an inverse association between serum levels of folate and VitB12 and adiposity. Obese children exhibited reduced serum levels of these micronutrients indicating their potential role in childhood obesity. This has implications for the development of preventive measures to reduce obesity risk in children [87]. An inverse correlation of VitB12 serum levels and body mass index was reported in obese women but not with insulin resistance [88]. The role of micronutrients was also investigated in other mental health disorders such as depression. Methyl-donor micronutrients contribute to SAM formation, which subsequently affects gene expression through methylation, helping to normalize homocysteine levels and prevent oxidative damage commonly observed in patients with major depressive disorder [89]. Low plasma levels and low dietary intake of folate and VitB12 were reported in individuals aged > 18 years with major depressive disorders [90].

Interestingly, an Australian study found no association between the postnatal intake of methyl-donor nutrients such as methionine, folate, VitB2, VitB6, VitB12 and global DNA methylation levels in the buccal cells of children (N = 73) at the age of four years. However, the study revealed higher DNA methylation levels in males, suggesting a gender-specific difference in methylation. To gain more insightful results, it might be beneficial to quantify DNA methylation for specific biomarkers in the blood and conduct the study with a larger cohort of children [91].

### 3.4. Methionine and SAM

Methionine is also a nutrient that can be derived from food such as meat, fish, dairy products, eggs, nuts, seeds, and legumes. It acts as a methyl-donor micronutrient in one-carbon metabolism. SAM is a product of methionine which is generated via the activity of methionine adenosyl methyltransferase (MAT). SAM is the main methyl-donor that methylates several substrates and participates in biological processes such as methylation and maintaining the integrity of cellular membranes via the formation of phosphatidylcholine. SAM can be converted to SAH which can be remethylated to regenerate methionine by donating a methyl group from 5-methyltetrahydrofolate (5-MTHFR) mediated by methionine synthase in a VitB12 -dependent manner. So, any disruption in the methionine cycle, closely intertwined with the folate cycle, can significantly impact various biological processes (Figure 1). The dysregulation of SAM levels has been linked to the etiology of many neuropsychiatric disorders [92]. For example, elevated homocysteine levels serve as an indicator of reduced folate, VitB12 and SAM levels. These findings are often observed in patients with depression, reinforcing the connection between micronutrients levels and mental health disorders. This implies that maintaining optimal levels of these micronutrients could potentially enhance health outcomes for individuals with depression [93].

SAM, which is formed from methionine, is also involved in the synthesis of monoamine neurotransmitters such as norepinephrine, dopamine, and serotonin. These neurotransmitters are synthesized by enzymes that take the methyl group from SAM for their enzymatic activity and are implicated in regulating mood and emotion beside other functions in the brain [94,95]. Although the relationship between these monoamine neurotransmitters and mood disorders is complex, it is suggested that SAM may have antidepressant-like effects by influencing the synthesis of these neurotransmitters [96]. In individuals with major depressive disorder (MDD), elevated plasma homocysteine levels are observed alongside reduced serum and cerebrospinal fluid (CSF) levels of SAM and folate, as well as decreased CSF levels of the monoamines dopamine, norepinephrine and serotonin [97]. An animal study showed that L-methionine supplementation, a precursor of SAM, reversed the epigenetic programming of early stress in adult offspring. Maternal L-methionine supplementation normalized the stress response in rat offspring by rectifying the methylation of the nerve growth factor inducible protein A (NGFA) binding site on the glucocorticoid receptor (GR) promoter [98]. Furthermore, clinical improvements were reported in MDD patients when SAM was used as a supplement for treatment, though these findings should be interpreted with caution [99].

Dysregulation of one-carbon metabolism has been reported in schizophrenia, with reduced activity of MAT, a crucial enzyme for SAM formation, noted in the erythrocytes of schizophrenics [100]. Given that decreased SAM levels can influence homocysteine levels, a nested case–control human study within a large birth cohort was conducted. This study revealed a connection between elevated maternal serum homocysteine levels during the third trimester of pregnancy and an increased risk of schizophrenia in the offspring during adulthood [101]. The heightened risk of schizophrenia in the offspring in response to dysregulation in micronutrient levels during the prenatal period was proposed to be mediated by epigenetic mechanisms [102]. For instance, the levels of DNA methyltransferases such as DNMT1 in telencephalic GABAergic neurons and DNMT3a in layers I and II of the cortex were both elevated in schizophrenic brains. Moreover, an increased expression of DNMT1 but to lesser extent DNMT3a was also reported in peripheral blood lymphocytes of these individuals, suggesting that peripheral blood could be used to detect epigenetic markers in schizophrenia [103]. Hypermethylation of the reelin gene promoter, a protein that is needed for memory formation, neuronal migration, axonal branching, and synaptogenesis was also detected in schizophrenic postmortem brains, with a significant reduction in its expression and activity. The hypermethylation was detected in the CpG islands flanking two regulatory sites in the reelin promoter, the cyclic AMP response element (CRE) and a stimulating protein-1 (SP-1) sites [104].

The role of methyl-donor micronutrients supplementation on brain programming by early life stress (ES) was investigated in mice. Maternal supplementation of these micronutrients such as methionine and B vitamins from postnatal day (PD2-PD9) restored methionine levels in the plasma and brain of the offspring and improved ES-induced cognitive impairments as detected using behavioral tests in the offspring. These changes were associated with the normalization of plasma corticosterone levels in the offspring implicating a normalization of the stress HPA axis response. Interestingly, this study reported that early life stress or maternal supplementation with methyl-donor micronutrients did not change the methylation of GR by epigenetic mechanisms in the mice hippocampus as was reported in other studies, indicating that GR expression change by early life stress is a species-specific effect and could be due to methodological differences [105]. Table 1 summarizes the correlation between methyl-donor micronutrients and both metabolic and mental health disorders.

## 4. Recommendation and Future Considerations

The connection between early life nutrition and mental health outcomes is increasingly recognized. Mental well-being is vital for longevity, productivity, and reducing healthcare costs. While mental health disorders can be influenced by genetics, environmental factors like eating and lifestyle habits play a significant role. However, the relationship between early nutritional interventions and lifelong mental health is correlational, not causal. This may be due to individuals’ reluctance to share dietary information, lack of public education on nutrition’s benefits for mental health, and inadequate funding for research. Challenges in translating animal model studies to human applications, individual dietary response variations, and environmental factors complicate this field. Future research should focus on understanding how methyl-donor micronutrients influence genes related to metabolic and stress-related pathways, aiming to improve health outcomes and prevent diseases.

Technological progress has improved our ability to personalize services through data analysis, notably in healthcare. Health professionals and dietitians ought to collaborate with patients to understand their dietary habits, enabling the creation of a customized nutritional plan at an early age that reflects everyone’s unique needs and preferences. Such personalized approaches have the potential to prevent, manage, or mitigate various health conditions later in life. Understanding how micronutrients influence gene regulation is crucial for developing effective nutritional strategies at early stages of life that not only enhance life quality and mental well-being but also prevent diseases like cardiovascular disease, diabetes, and obesity.

## 5. Conclusions

Methyl-donor micronutrients, which are derived from diet, are critical players in the functionality of one-carbon metabolism, a series of chemical reactions that regulate essential biological processes. These processes include neurotransmitter synthesis, the maintenance of cellular membrane integrity, nucleotide synthesis, methylation reactions, and the transsulfuration pathways. There is a notable association between nutritional status and the functionality of one-carbon metabolism, with evidence suggesting that nutrition influences the onset of mental health disorders. Studies have proposed that epigenetic mechanisms may underpin the connection between nutrition, genes, and both metabolic and mental health conditions. Despite a limited understanding of the effects of diet on the epigenome, it is known that dietary methyl-donor micronutrients regulate methylation pathways and the synthesis of neurotransmitters that regulate mood, emotion, and cognitive functions. Those findings presented in this review regarding the impact of these micronutrients on the brain and metabolism in rodent models underscore the necessity for more human-based research to confirm these findings. Additionally, we raise the question of whether the mechanisms of epigenetic regulation associated with mental and metabolic disorders differ by sex, a topic that merits elucidation in future studies. Our current understanding of how methyl-donor micronutrients influence epigenetic mechanisms, particularly in humans, is limited. Therefore, delineating the interplay between nutrition, micronutrients, epigenetic mechanisms, and brain health in humans is crucial. Such knowledge could enable pediatricians and dietitians to implement more personalized interventions early in life, aiming to prevent childhood obesity and reduce the risk of developing metabolic and mental health disorders later in life.

## Figures and Tables

**Figure 1 ijms-25-04036-f001:**
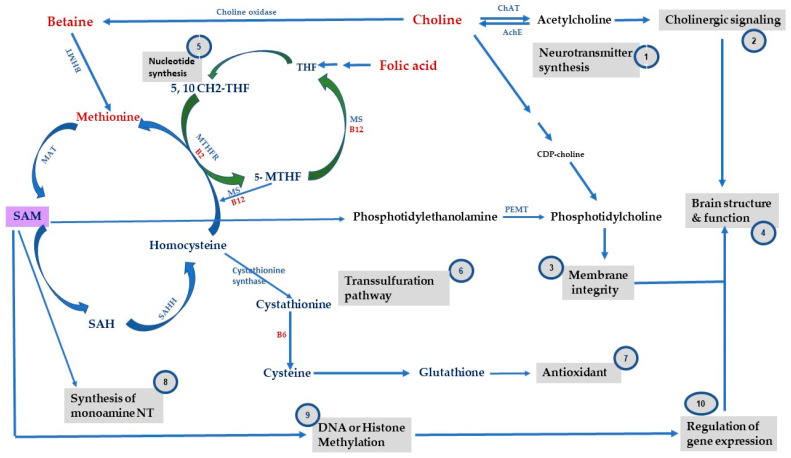
**One-carbon metabolism and methyl-donor micronutrients.** This figure shows the main components of one-carbon metabolism, folate cycle, methionine cycle, transsulfuration pathway and the contribution of micronutrients such as choline, betaine, folate, methionine, VitB2, VitB6, and VitB12 in several biological processes. Biological processes are highlighted in gray. Micronutrients are highlighted in red. Enzymes are highlighted in light blue (THF = tetrahydrofolate, SAM = S-adenosylmethionine, SAH = S-adenosylhomocysteine, SAHH = SAH hydrolase, BHMT = betaine homocysteine methyltransferase, PEMT = phosphotidylethanolaminetransferase, MAT = methionine acetyltransferase, MS = methionine synthase, MTHFR = methyltetrahydrofolate reductase, ChAT = choline acetyltransferase, AchE = acetylcholine esterase. Adopted from Bekdash, 2023 and slightly modified [22].

**Table 1 ijms-25-04036-t001:** The link between methyl-donor micronutrients and metabolic and mental health disorders.

Methyl-Donor Micronutrients	Condition	Effects	References
**Folate**	Maternal supplementation with folate from GD13-GD20 in rats	Normalized Let-7a and miR-34 levels at GD20 and normalized the expression of genes related to embryonic development, cell migration, axonal guidance, and vesicular trafficking	[33]
	Deficiency in folate and VitB12 in rats	Altered expression of metabolic genes in the rat hypothalamus	[30]
	Folate deficiency in mice from GD11-GD17	Decrease in NSC proliferation and differentiation and increase in neuronal apoptosis in fetal mouse brain	[52]
	Supplementation of high folate diet to Wistar rat offspring	Decrease in *Pomc* methylation and improved glucose response of offspring to glucose load	[55]
	Folate Supplementation in neonatal isolation rat model	Hypomethylation of BDNF and increase in its expression + mitigated autistic and anxiety-like behavior in offspring	[56]
	Supplementation with high methyl-vitamins gestational diet (B12, B6, folate) in Wistar rats	Hypermethylation of hypothalamic *Pomc* and *5-Htr2a* + leptin and insulin resistance in offspring	[36]
	High maternal folate and low VitB12 in pregnant women	Insulin resistance and higher adiposity in children at age 6	[57]
	High maternal folate but not VitB12 in pregnant women	Insulin resistance in children at age of 5, 9.5, and 13.5	[58]
	High maternal folate and low VitB12 in pregnant women	Insulin resistance and higher adiposity in children at age 6	[60]
	Low dietary intake of folate in obese females	Insulin resistance and decrease in methylation of metabolic-related gene *Camkk2* in WBCs	[62]
**Choline**	Prenatal choline supplementation from GD11-GD18 in iron-deficient rats	Mitigated the effects of iron deficiency on hippocampal genes associated with autism and schizophrenia	[32]
	High maternal choline intake in Wistar rats	Increase in hypothalamic expression of *Npy* in pups at birth + higher food intake and body weight gain postweaning in offspring	[69]
	Low choline intake from GD12-GD17 in rodents	Disruption of fetal progenitor cell differentiation and changes in methylation	[70]
	Paternal supplementation of FMCD diet in F0 mice	Anxiety-like behavior and impairment in memory consolidation with modest PP1c methylation in male F1 offspring	[77]
	Maternal choline intake in women with gestational diabetes	Cord blood CRH methylation and decrease in plasma cortisol levels	[61]
	Low choline levels in plasma in the Hardland Health Study	Anxiety in adults	[73]
	Low plasma choline at 16-week gestation in women	Dysregulated cerebral P50 inhibition in infant Increased predisposition to mental illness	[74]
**VitB12**	Increase intake of VitB12 and B6 in humans	Reduced depressive-like symptoms in adults	[85]
	Intake of folate and VitB12 in humans	Improved symptoms in schizophrenics	[86]
	Reduced folate and VitB12 levels in children	Associated with obesity in Mexican American children	[87]
	Low dietary intake of folate and B12 in humans	Associated with MDD individuals	[89]
**Methionine/SAM**	SAM supplementation in rats prior to a forced swimming test	Increase in methylation of immediate early genes with elevated levels of Dnmt3a in DG granule neurons and reduced stress response	[78]
	Maternal L-methionine supplementation in rats	Normalized methylation of NGFA binding site on GR promoter in offspring	[98]
	Supplementation of methionine and B vitamins from PD2-PD9 in mice subjected to early life stress (ES)	Prevented ES-induced rise in corticosterone and rescued ES-induced cognitive impairments in offspring	[105]
	Elevated plasma homocysteine levels and reduced levels of SAM, folate, and monoamine metabolites in CSF of patients	Observed in MDD patients	[97]
	SAM supplementation in MDD patients	Clinical improvements in MDD patients	[99]
	Elevated homocysteine levels during third trimester of pregnancy	Increased risk of schizophrenia in offspring	[101]

In this table, select animal studies then select human studies are summarized for each methyl-donor micronutrient.

## Data Availability

Not applicable.

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
