# Peer review of "Epigenetics, Nutrition, and the Brain: Improving Mental Health through Diet"

_ijms, 2024, doi:10.3390/ijms25074036_

Round 1

Reviewer 1 Report

Comments and Suggestions for Authors

The thematic proposed is interesting but it should be better covered:

Line 254 One paper can not justify the sentence, particularly folic acid is normally supplemented, and no mechanism is proposed.

In general, it should be distinguished between reports operated in humans and animals, grouping them in two different boxes; studying animals could be just a first step, but non-conclusive.

As well as just one study on humans can not be as direction.

And for every mechanism should be outlined

Even not being related to metil donation, other nutrients should be included: creatine (10.2174/0118761429272915231122112748) omega 3 10.3390/nu13041074), beta-carotene (10.3390/brainsci13101468)

having a fundamental role in brain function and development

A practical recommendation could be useful

Comments on the Quality of English Language

It needs revision

Author Response

Reviewer # 1:

The thematic proposed is interesting but it should be better covered:

  • Line 254 One paper can not justify the sentence, particularly folic acid is normally supplemented, and no mechanism is proposed.

I added more references to support this statement, References 59, 60, 61 & 62 (Line 277). I also summarized these references in Table 2 (Line 497).

  • In general, it should be distinguished between reports operated in humans and animals, grouping them in two different boxes; studying animals could be just a first step, but non-conclusive.

I revised Table 2 to distinguish between animal studies and human studies. Please refer to line 497. I included a footnote on line 498. I also rearranged the order of references to make the table clearer.

  • As well as just one study on humans cannot be as direction.

In the manuscript, I included several animal and human studies. These studies were also summarized in Table 2. I also added references 60 and 62.

  • And for every mechanism should be outlined.

The mechanism is stated when applicable for each study in this paper.  For references 59, 60, 61 & 62, these studies did not include the mechanism.  These studies assessed the levels of micronutrients during pregnancy on insulin resistance.

Please note that I am focusing on changes in epigenetic mechanisms such as DNA methylation, histone modifications or microRNAs. These mechanisms are stated in this manuscript, when applicable.

  • Even not being related to metil donation, other nutrients should be included: creatine (10.2174/0118761429272915231122112748) omega 3 3390/nu13041074), beta-carotene (10.3390/brainsci13101468), having a fundamental role in brain function and development
    I agree with the reviewer that creatine, omega 3 and beta carotene may play a role in cognitive function. However, the focus of this paper is on the role of methyl-donor micronutrients that cause changes in gene expression by epigenetic mechanisms and are linked to mental health or metabolic problems. This is clearly stated in the manuscript (Lines 100 -103). Adding other nutrients will derail the focus of the paper. Also, creatine, and omega-3 are not considered micronutrients. Discussing these nutrients and their role in cognitive function would be the focus of another manuscript.

  • A practical recommendation could be useful
    A section entitled “Recommendation and Future Considerations” section is now added (Line 500).

Reviewer 2 Report

Comments and Suggestions for Authors

The work is devoted to a review of articles focusing on the effects of consumed methyl-donor micronutrients on the brain. The text is well written and consistent. However, there is some confusion in the list of references. The author also has a similar review (doi:10.3390/ijms24032346), it should be well explained how this work differs from the previous one. Once corrections have been made, the article can be published.

The comments include requirements and recommendations.

1) It would be nice if in the Abstract and Introduction you could clearly state the purpose of the review, what you want to achieve by reviewing the works you have selected.

2) Bring some items from References ([16] (line 81), [27] (lines 561-562), [29] (lines 565-566), [37] (lines 588-589)) into a more acceptable format

(eg ‘Interplay between Metabolism and Epigenetics: A Nuclear Adaptation to Environmental Changes - PubMed. https://pubmed.ncbi.nlm.nih.gov/27259202/ (accessed 2024-03-06).’

=>

Etchegaray, J. P.; Mostoslavsky, R. Interplay between Metabolism and Epigenetics: A Nuclear Adaptation to Environmental Changes. Mol Cell 2016, 62(5), 695-711. https://doi.org/10.1016/j.molcel.2016.05.029.)

3) Figure 1 (line 134) should be next to the first mention in the text (line 105).

4) ‘Increasing dietary methyl-donor content in the high anxiety group improved anxiety and depressive-like behavioral phenotypes which were demonstrated as decreased in immobility in the forced swim test [34]’ (lines 184-187):

The reference does not match the statement. Check the order of work in References.

5) Starting from reference [87] (lines 385, 721), the order of works in References has been shifted. Check all the references for consistency with the text.

6) How your previous review (doi:10.3390/ijms24032346) is fundamentally different from the current one? Emphasize this in the text, explain to readers why these are different works.

Author Response

Reviewer # 2

The work is devoted to a review of articles focusing on the effects of consumed methyl-donor micronutrients on the brain. The text is well written and consistent. However, there is some confusion in the list of references. The author also has a similar review (doi:10.3390/ijms24032346), it should be well explained how this work differs from the previous one. Once corrections have been made, the article can be published.

 The comments include requirements and recommendations.

  • It would be nice if in the Abstractand Introduction you could clearly state the purpose of the review, what you want to achieve by reviewing the works you have selected.

I added a statement in the abstract since I am restricted to 200 words (Lines 20-22).  I added a small paragraph at the end of the Introduction (Lines 66-69 and Lines 76-78). I also included “Recommendations & Future Considerations” section (Line 500).

  • Bring some items from References([16] (line 81), [27] (lines 561-562), [29] (lines 565-566), [37] (lines 588-589)) into a more acceptable format.

I followed the ACS style as recommended by IJMS.  If I made a mistake and the ACS style is changed, I would like the journal’s help in fixing it, if possible.  I really do not know how to change it using my Zotero.

  • Figure 1 (line 134) should be next to the first mention in the text (line 105).

Figure 1 is now on line 114. 

4) ‘Increasing dietary methyl-donor content in the high anxiety group improved anxiety and depressive-like behavioral phenotypes which were demonstrated as decreased in immobility in the forced swim test [34]’ (lines 184-187):

Thanks for bringing this to my attention. I checked all my citations.  All are now fixed.

5) Starting from reference [87] (lines 385, 721), the order of works in References has been shifted. Check all the references for consistency with the text.

 All citations are now fixed and consistent with the text.  There was a shift in one reference which affected the rest.

6) How your previous review (doi:10.3390/ijms24032346) is fundamentally different from the current one? Emphasize this in the text, explain to readers why these are different works.

I added a statement in the abstract (Lines 20-22) and made minor changes at the end of the Introduction (Lines 66-69 and Lines 76-78). I also included “Recommendations & Future Considerations” section (Line 500).

Round 2

Reviewer 1 Report

Comments and Suggestions for Authors

The author improved the manuscript according to my request.

Just a clarification: if creatine is not to be seen as a micronutrient, why should it be choline or betaine?

Comments on the Quality of English Language

Just some more, but overall ok

Author Response

Reply to Reviewer # 1 – Round 2

March 30th, 2024

Dear Reviewer # 1,

Thank you very much for your valuable comments. Your feedback enabled me to revise my manuscript and make it better.

Related to your comment about creatine, this manuscript emphasizes the role of methyl-donor micronutrients that play key roles in the one-carbon metabolism and contribute to SAM formation.  SAM is the universal methyl-donor critical for histone and DNA methylation, which are two main epigenetic mechanisms.

The determination of whether a substance is classified as a micronutrient depends on the body’s ability to synthesize it in sufficient amounts to maintain human health.   Choline and Betaine have essential roles that cannot be fully met by endogenous synthesis alone.  They are synthesized in small amounts in the body (I included this statement on line 60) necessitating their intake through diet, which underpins their status as micronutrients.

As for creatine, while beneficial and possibly essential for optimal muscle and brain function under specific conditions, the body can synthesize it in sufficient quantities from amino acids and can also obtain it from diet.  Unlike choline and betaine, the body synthesizes enough creatine to meet its basic physiological needs. That is why creatine cannot be considered a micronutrient.

Moreover, little is understood about the potential impact of creatine on epigenetic mechanisms in the context of metabolism and mental health problems, the focus of this paper.

Thank you

Rola Bekdash